# Cancer Essential Genes Stratified Lung Adenocarcinoma Patients with Distinct Survival Outcomes and Identified a Subgroup from the Terminal Respiratory Unit Type with Different Proliferative Signatures in Multiple Cohorts

**DOI:** 10.3390/cancers13092128

**Published:** 2021-04-28

**Authors:** Kuo-Hao Ho, Tzu-Wen Huang, Ann-Jeng Liu, Chwen-Ming Shih, Ku-Chung Chen

**Affiliations:** 1Graduate Institute of Medical Sciences, College of Medicine, Taipei Medical University, Taipei 11031, Taiwan; d119106001@tmu.edu.tw (K.-H.H.); tw.huang@tmu.edu.tw (T.-W.H.); 2Department of Biochemistry and Molecular Cell Biology, School of Medicine, College of Medicine, Taipei Medical University, Taipei 11031, Taiwan; 3Department of Microbiology and Immunology, School of Medicine, College of Medicine, Taipei Medical University, Taipei 11031, Taiwan; 4Department of Neurosurgery, Taipei City Hospital Ren-Ai Branch, Taipei 10629, Taiwan; DAB90@tpech.gov.tw

**Keywords:** lung adenocarcinoma (LUAD), terminal respiratory unit (TRU), TP53 mutation, E2F

## Abstract

**Simple Summary:**

Several genes are essential for tumor growth and predict poor prognoses of patients, however, most of their roles in lung adenocarcinoma (LUAD) are unclear. In addition, a good classification strategy for cancer patients would be a useful tool for future personalized medicine. In LUAD, the existing subtype classification, including the terminal respiratory unit (TRU), proximal-inflammatory (PI), and proximal-proliferative (PP) subtypes, is mainly based on genes with variant expression levels across patients without considering the oncogenetic roles of those genes. Thus, the LUAD essential genes were identified and used to stratify patients into distinct survival outcomes, TP53 mutation statuses, E2F target activities, and tumor mutation burdens. Moreover, TRU-type patients could be further divided into clinically and molecularly different subgroups based on our classifier. Integration of existing subtypes with our classification strategy provides a more comprehensive understanding of the heterogeneity of LUAD, and can guide us to identifying potential targets for future personalized medicine.

**Abstract:**

*Background*: Heterogeneous features of lung adenocarcinoma (LUAD) are used to stratify patients into terminal respiratory unit (TRU), proximal-proliferative (PP), and proximal-inflammatory (PI) subtypes. A more-accurate subtype classification would be helpful for future personalized medicine. However, these stratifications are based on genes with variant expression levels without considering their tumor-promoting roles. We attempted to identify cancer essential genes for LUAD stratification and their clinical and biological differences. *Methods*: Essential genes in LUAD were identified using genome-scale CRIPSR screening of RNA sequencing data from Project Achilles and The Cancer Genome Atlas (TCGA). Patients were stratified using consensus clustering. Survival outcomes, genomic alterations, signaling activities, and immune profiles within clusters were investigated using other independent cohorts. *Findings*: Thirty-six genes were identified as essential to LUAD, and there were used for stratification. Essential gene-classified clusters exhibited distinct survival rates and proliferation signatures across six cohorts. The cluster with the worst prognosis exhibited TP53 mutations, high E2F target activities, and high tumor mutation burdens, and harbored tumors vulnerable to topoisomerase I and poly(ADP ribose) polymerase inhibitors. TRU-type patients could be divided into clinically and molecularly different subgroups based on these essential genes. *Conclusions*: Our study showed that essential genes to LUAD not only defined patients with different survival rates, but also refined preexisting subtypes.

## 1. Introduction

Lung cancer, a leading cause of cancer-associated mortality, can be categorized into small-cell lung carcinoma (SCLC) and non-SCLC (NSCLC). Lung adenocarcinoma (LUAD) is the major subtype of NSCLC which accounts for around 40% of all lung cancer patients [1]. Although various drugs for treating LUAD have been extensively investigated, survival rates of LUAD patients have still not dramatically improved [2]. Therapeutic strategies for LUAD are based on histopathological features or tumors presenting with targetable genomic alterations like *epidermal growth factor receptor 1* (*EGFR**1*) mutations [3], or translocation of *ALK, RET,* or *ROS1* [4,5,6]. However, these factors still do not completely capture the highly heterogeneous features of LUAD. Thus, it is necessary to characterize this complex disease using more-sophisticated approaches.

With the development of high-throughput sequencing technology, several studies have tried to define LUAD based on transcriptome profiles. Initially, LUAD was stratified into bronchoid, magnoid, and squamoid subtypes with significant clinical differences such as stage-specific survival [7]. Through large-scale multiomics investigations of LUAD by The Cancer Genome Atlas (TCGA), these subtypes were renamed terminal respiratory unit (TRU), proximal-proliferative (PP), and proximal-inflammatory (PI) subtypes [8]. Among these types, TRU has favorable prognoses and harbors tumors presenting *EGFR* mutations. The PP and PI subtypes have poorer survival outcomes. The former harbors *KRAS* mutations and inactivation of *STK11*, while the latter has co-mutations of *NF1* and *TP53*. However, stratification of these subtypes is mainly based on gene candidates with highly variant expressions across tumor samples without considering the roles of these candidates in tumor malignancy. Thus, we attempted to identify a subset of cancer essential genes to reclassify LUAD patients.

Project Achilles uses a genome-scale CRISPR-Cas9 tool to individually knock out each gene, thereby identifying candidates which are critical for cancer survival [9]. Taking advantage of Project Achilles and RNA sequencing (RNA-Seq) data from LUAD patients, we were able to pinpoint essential genes responsible for LUAD malignancy. These essential genes were used to classify LUAD patients into different molecular types. Clinical differences of these molecular types in multiple cohorts were investigated. Additionally, a new subset of patients with distinct prognoses in the TRU subtype was identified using our classification. These findings may allow us to refine the preexisting subtype classification of LUAD, and also guide us in identifying tumors that may be vulnerable to specific treatments.

## 2. Materials and Methods

### 2.1. Retrieving LUAD Patient Data and Identifying Essential LUAD Genes

RNA Seq data and clinical information of TCGA LUAD patients (*n* = 511) were retrieved from UCSC Xena (https://xena.ucsc.edu/; 20 July 2019). Other LUAD datasets were obtained from Gene Expression Omnibus (GEO) datasets. Expression data from TCGA and GSE140343 (*n* = 51) were normalized as fragments per kilobase of transcripts of million mapped reads (FPKM) and then log2 transformed. The GSE68465 (*n* = 432), GSE72094 (*n* = 398), GSE50081 (*n* = 127), and GSE31210 (*n* = 226) datasets consisted of microarray data, and their expressions were normalized by robust multichip averaging with log2 transformation. Genome-wide CRISPR screening of LUAD cells was downloaded from the DepMap portal (https://depmap.org/portal/download/; 20 July 2019). Dependency scores for around 17,000 candidate genes were calculated using the CERES algorithm [9]. Candidate genes were defined as essential genes with a CERES score of <−1 across 75% of LUAD cell lines (*n* = 31). Using the limma package, differentially expressed gene (DEG) analyses were conducted between tumor and paired normal tissues from TCGA and GSE140343 RNA Seq data. Candidates with a false discovery rate (FDR) of <0.01 and a multiple of change of >1 were considered to be significantly upregulated in tumor tissues.

### 2.2. Classification of LUAD Subtypes and Clinical Feature Analysis

To identify RNA expression subtypes of LUAD, i.e., TRU, PP, and PI, in a previous study [8], the subtypes were assigned to each LUAD tumor using the nearest centroid predictor. A Pearson correlation analysis was performed to correlate expression profiles of each tumor with the nearest centroid predictor. The subtype of each patient was predicted based on the maximum correlation coefficient. Essential genes we identified were used to perform an unsupervised consensus clustering with TCGA LUAD data, and a partitioning around medoids (PAM) clustering algorithm was used. One thousand permutations with a 0.95 random fraction of essential genes in each iteration were repeated to perform the clustering analyses. Judging from the delta area plot, the optimal cluster was selected according to whether no appreciable increase was present. Expressions of essential genes in TCGA were median centered, and they were used to develop the nearest centroid classifier that could predict essential gene-classified clusters. The classifier was applied to the GEO and GDSC datasets to predict subtypes of LUAD. Expressions of essential genes are shown in a heatmap. To compare 5-year overall survival in each subtype, a log-rank test was performed. A multivariate Cox regression was conducted to investigate whether the essential gene-classified cluster was an independent prognostic factor considering tumor stages. To evaluate the relation between essential-classified clusters and tumor stage, a logistic regression analysis was performed. The dependent variable was tumor stage, and we divided it into binary variable. The patients in TCGA, GSE140343, GSE68645 and GSE72094 were categorized into high stage (stage III and stage IV) and low stage (stage II and stage I) groups. For the low stage patients in GSE50081 and GSE31210, they were divided into stage II and stage I. According to previous studies [10,11,12], four pathological subtypes with distinct survival have been identified including lepidic predominant non-mucinous adenocarcinomas (ADCs), acinar/papillary predominant non-mucinous ADCs, micropapillary/solid predominant non-mucinous ADCs, and invasive mucinous ADCs. A Fisher exact test was performed to compare the enrichment for a certain essential gene-stratified cluster in a given pathological type.

### 2.3. Comparisons of Pathway and Transcription Factor (TF) Activities in LUAD Patients

To compare signaling pathways that differed within essential gene-classified clusters, a single-sample gene set enrichment analysis (ssGSEA) was performed to evaluate the degree of activation of Hallmark pathways [13]. TF activity was calculated according to Garcia-Alonso et al. [14]. Briefly, a group of high-confidence human TFs and their target genes were defined based on TF-binding site predictions, text mining-derived TF-target interactions, and chromatin immunoprecipitation coupled with high-throughput (ChIP-X) data. Transcriptome profiles of these TF targets were used to infer TF activities of each patient by performing analytical rank-based enrichment analyses (aREAs). The Kruskal-Wallis test with post-hoc Dunn’s test was conducted to identify the top signaling pathways or TFs that significantly differed within essential gene-stratified clusters (*p* < 10^−3^). A multivariate linear regression was conducted to evaluate the association between ssGSEA-inferred E2F targets activity and essential gene-classified clusters considering tumor stage as a covariate.

### 2.4. Copy Number Variations and Mutation Analyses

Copy number variations and genomic mutation data of TCGA LUAD were downloaded from UCSC Xena (https://xena.ucsc.edu/; 20 July 2019). Copy number segment data were analyzed using GISTIC 2.0 to identify significant focal copy number alterations in LUAD patents. Mutation data were generated by the Multi-Center Mutation Calling in Multiple Cancers (MC3) project. Genes were divided into binary calls as either a non-silent mutation or wild type (WT). Fisher’s exact test was carried out to explore changes in genomic mutations and copy numbers that were significantly enriched in specific essential gene-classified clusters (*p* < 10^−3^).

### 2.5. Drug Discovery in Essential Genes-Stratified Clusters

To uncover the drug candidates that exhibited different efficacy in essential gene-classified clusters, we queried the genomics of drug sensitivity in cancer (GDSC) database (https://www.cancerrxgene.org; 16 January 2020). This database contains nearly 1000 genetically characterized human cancer cell lines treated with an array of anti-cancer therapeutics (367 compounds). Among these cell lines, 46 of them belongs to LUAD. We applied our KNN prediction to classify these cell lines into our essential genes-stratified clusters and compared the differences of the area under the receiver operator characteristics curve (AUC) drug responses among these clusters. Only 257 compounds which were tested on more than 75% of the LUAD cell lines were selected to analyze. The Kruskal-Wallis test with a *p* value < 0.05 was considered as significant difference.

### 2.6. Evaluating the Tumor Mutation Burden (TMB) and Immune Infiltration in Essential Gene-Classified Clusters

The TMB was derived from the sum of gene-coding errors, and base substitution insertions or deletions divided by the length of the human exon (38 MB) [15]. The degree of immune infiltration was measured using the method described in ESTIMATE. Briefly, an ssGSEA using RNA expression signatures related to immune cells was conducted to infer immune infiltration in tumors [16]. This method was applied to GEO and TCGA LUAD data to estimate the degree of immune infiltration. The TMB and immune infiltration were compared in essential gene-stratified clusters by the Kruskal-Wallis test with post-hoc Dunn’s test.

## 3. Results

### 3.1. Identification of Essential Genes for Promoting LUAD Malignancy

An analysis pipeline was designed to identify and characterize subtypes, clinical features, and molecular profiles of LUAD patients in the present study (Figure 1A). To pinpoint crucial gene candidates responsible for LUAD malignancy, we investigated genome-wide CRISPR-based loss-of-function screens derived from DepMap. In total, 693 genes were found to be crucial for maintaining survival in 31 LUAD cell lines (Appendix A). To identify which candidates among these 693 genes were aberrantly expressed in tumor tissues, DEG analyses were carried out to compare tumor tissues with paired normal tissues in TCGA (number of pairs = 59) and GSE140343 (number of pairs = 49) RNA Seq data. Thirty-six of 693 essential genes were significantly upregulated in tumor tissues (with a multiple of change of >1 and an FDR of <0.01) (Figure 1B, Appendix A). Gene expression correlation analyses indicated that these 36 candidate genes were strongly associated with each other in both TCGA (median Pearson’s *R* = 0.625) and GSE140343 (median Pearson’s *R* = 0.621) (Figure 1C and Appendix A). This finding implied that cancer essential genes were coordinately modulated by common regulators. Functionally, pathway analyses revealed that these 36 LUAD essential genes were enriched in cell proliferative signaling pathways including DNA replication checkpoint, transcription involving the G_1_/S transition, and DNA replication initiation (Figure 1D).

### 3.2. Essential Genes Stratified LUAD Patients with Different Prognoses and E2F Signaling Activities

Unsupervised consensus clustering was performed to classify TCGA LUAD patients into three robust clusters using the 36 identified essential genes (Figure 2A,B). A log-rank test demonstrated that these three clusters exhibited significantly different prognoses (log-rank test *p* = 0.0039) (Figure 2C). Clusters 1, 2, and 3 respectively exhibited the worst prognosis, a moderate survival time, and the most favorable prognosis. The associations between tumor stages and essential genes-stratified clusters were analyzed by conducting logistic regression. The cluster 1 patients were significantly associated with higher tumor stage comparing with cluster 3 (log of the odds ratio = 0.99, *p* value < 0.01) (Appendix A). Besides tumor stages, the associations between pathological types and clusters were also analyzed. We identified that cluster 3, cluster 2, and cluster 1 patients were respectively enriched in lepidic (odds ratio = 32.86, *p* value < 0.01), acinar/papillary (odds ratio = 1.96, *p* value = 0.02), and micropapillary/solid ADCs (odds ratio = 5.35, *p* value < 0.01) (Appendix A). A multivariate Cox regression analysis indicated that the essential gene-stratified subtype was a prognostic factor independent of the lung cancer stage in TCGA LUAD patients (cluster 1 vs. cluster 3; hazard ratio (HR) = 1.6, *p* = 0.039) (Figure 2D). Expression levels of essential LUAD genes subsequently increased in the order of clusters 3, 2, and 1 (Figure 2E). To identify central regulators associated with distinct expression profiles of LUAD essential genes, we compared genomic alterations among these three clusters. Genomic mutation and copy number variation analyses revealed that the *TP53* mutation and chromosome 3q26.2 amplification were significantly enriched in cluster 1 (Figure 2F,G). The ssGSEA identified that E2F targets, the G_2_M checkpoint, and mitotic signaling were significantly upregulated in cluster 1 (Figure 2H, upper panel). Additionally, an aREA was performed to infer TF activities in LUAD patients. Activities of multiple E2F TFs, including E2F2, E2F3, and E2F4, were significantly higher in cluster 1 (Figure 2H, lower panel). A multivariate linear regression analysis demonstrated that the association between E2F targets and essential gene-classified clusters was independent of tumor stage (Appendix A). Combining findings of the pathway and transcription factor analyses, these clusters exhibited distinct E2F activities. Thus, these three clusters were denoted as having high (cluster 1), medium (cluster 2), and low (cluster 3) E2F activities.

### 3.3. Essential Genes Identify a New Subgroup from TRU-Type Patients with a Favorable Prognosis

A previous study divided LUAD patients into different molecular subtypes including TRU, PP, and PI based on 506 candidate gene expressions [8]. The TRU type had favorable prognoses, but PP and PI type patients had poor survival rates. The three essential gene-classified clusters were used to compare existing molecular subtypes (Figure 3A). Cluster 1 and 2 patients were highly enriched in PI (65.0% in cluster 1, 31.6% in cluster 2, and 3.4% in cluster 3) and PP (64.0% in cluster 1, 34% in cluster 2, and 2% in cluster 3). Cluster 3 patients were significantly enriched in TRU (8.1% in cluster 1, 43.8% in cluster 2, and 48.1% in cluster 3). Although both cluster 3 and TRU-type patients had favorable prognoses, nearly half (51.9%) of TRU patients contained cluster 1 and 2 types, which are poor survival groups. Therefore, we wondered whether essential gene-classified clusters could further subdivide TRU-type patients into distinct survival groups. By performing a log-rank test, TRU belonging to the cluster 3 subtype exhibited more-favorable prognoses compared to the other molecular types (Figure 3B). No survival differences were found among TRU without cluster 3, and PP and PI subtype patients. Additionally, we found that patients with lepidic pathological type were significantly enriched in TRU-cluster 3 (odds ratio = 36.86, *p* value < 0.001) (Appendix A). From these results, essential genes provide an additional characterization from preexisting molecular types of LUAD.

### 3.4. Validation of the Survival Significance of Essential Gene-Stratified Clusters in Multiple Cohorts

To validate the prominent roles of essential gene-classified clusters, TCGA LUAD patients were used as a training set to build a prediction model using the nearest centroid classifier. Because 14 of the 36 essential genes overlapped with candidates that stratified preexisting molecular types, only 22 genes remained for establishing the classification model (Appendix A). In addition, 13 of the 36 essential genes directly belonged to E2F targets (Appendix A). LUAD patients derived from GSE140343 (*n* = 51), GSE68465 (*n* = 432), GSE72094 (*n* = 398), and low-stage LUAD patients derived from GSE50081 (*n* = 127) and GSE31210 (*n* = 226) were divided into three clusters based on the classification model. Expressions of essential genes and E2F target signaling were obviously activated in the order of clusters 3, 2, and 1 (Appendix A). Logistic regression analysis demonstrated that cluster 1 patients were significantly associated with higher tumor stage comparing with cluster 3 in GSE50081 (log of odds ratio = 1.16, *p* value = 0.04) and GSE31210 (log of odds ratio = 2.05, *p* value < 0.01) (Appendix A). Additionally, GSE68465 contained the tumor grade information, and we identified that cluster 1 (log of the odd ratio = 2.16, *p* value < 0.01) and cluster 2 (log of the odd ratio = 2.89, *p* value < 0.01) patients associated with higher tumor grade compared with cluster 3 patients (Appendix A). Multivariate linear regression analysis showed that the clusters were correlated with E2F target activity independent of tumor stages in all the cohorts (Appendix A). Log-rank survival analyses demonstrated that cluster 1 patients had the significantly worst prognoses, while cluster 3 patients had favorable survival rates in the GSE68465, GSE72094, GSE50081, and GSE31210 datasets (Figure 4A). Although survival times did not significantly differ across essential gene-classified clusters in the GSE140343 dataset, their survival trends still followed a similar pattern as the other cohorts. The insignificant survival differences in the GSE140343 dataset might have been due to its small sample size (*n* = 51). Additionally, a multivariate Cox regression confirmed that essential gene-classified clusters acted as an independent prognostic factor considering tumor stages as covariates in the GSE68465, GSE72094, GSE50081, and GSE31210 datasets (Figure 4B). Similar to TCGA data, around half of TRU patients in these cohorts belonged to clusters 1 and 2 which exhibited high E2F signaling and proliferative signatures (Figure 4C). A survival analysis was also used to identify that TRU patients with cluster 3 genetic features exhibited better prognoses among different molecular types in three (GSE31210, GSE68465, and GSE50081) of the five datasets. These results suggested that essential LUAD genes were prominent molecular predicators that could identify LUAD patients with distinct proliferative signatures and prognoses. Importantly, this signature could be further used to refine preexisting RNA expression subtypes of LUAD.

### 3.5. Potential Drug Discovery and Immune Environment Characterization of Essential Gene-Classified Clusters

To identify drugs that exhibited distinct efficacy in essential gene-classified clusters, the prediction model was applied to the GDSC database (Figure 5A). Areas under the receiver operator characteristics curve (AUC) of drug responses within clusters were compared. Totally, the GDSC database contains 367 compounds. Only 257 compounds which were tested on more than 75% of the LUAD cell lines were used for analyses (Appendix A). AUCs of SN-38, a topoisomerase I inhibitor, and talazoparib, a poly(ADP ribose) polymerase (PARP) inhibitor, were significantly lower in cluster 1 LUAD cells (Figure 5B). This suggested that cluster 1 tumors with highly proliferative signatures were vulnerable to drugs targeting DNA-replication mechanisms. Understanding the immune microenvironment and mutation burden in tumors could guide us in identifying tumors sensitive to immunotherapies. Thus, we compared differences in these factors within essential gene-classified clusters in TCGA and GEO datasets. The ESTIMATE-derived immune infiltration score did not prominently differ within these clusters (Figure 5C and Appendix A), but the TMB subsequently decreased in the order of clusters 1, 2, and 3 (Figure 5D). Previously reported expression subtypes exhibited distinct immune profiles with the TRU and PI subtypes demonstrating higher immune infiltration compared to the PP subtype. Activities of E2F target signaling were lower in the TRU subtype (Figure 5E and Appendix A). Further, categorization of these previously reported subtypes with essential gene-stratified clusters led to distinct E2F signaling (Figure 5F and Appendix A). Herein, a combination of our essential gene-stratified clusters and previously reported expression subtypes more comprehensively captured proliferative and immune profiles of LUAD (Figure 5G). The PI type presented high immune infiltration and high proliferative features; the PP type exhibited low immune infiltration and high proliferative features; the TRU type was subdivided into high immune infiltration/high proliferative and high immune infiltration/low proliferative groups with distinct prognoses. In conclusion, E2F signaling-related essential genes could identify highly immune infiltrative TRU patients with distinct proliferative signatures and prognoses.

## 4. Discussion

Inter-patient diversity in LUAD indicates the importance of identifying genetically different subgroups with distinct survival and druggable targets. By combining Project Achilles and LUAD patient RNA Seq data, 36 cancer essential genes involved in cell proliferation pathways were nominated. Clinically, these essential gene signatures stratified LUAD patients into different survival groups in multiple cohorts. Molecularly, *TP53* mutations and chromosome 3q26.2 amplification were enriched in patients with the worst prognoses as identified by essential genes. Additionally, E2F targets and E2F transcription activities were activated in the group with the worst prognoses. GDSC drug analyses identified that the high E2F-signaling group was sensitive to cell proliferation inhibitors including talazoparib and SN-38. Intriguingly, essential gene-classified clusters further identified a group of TRU patients with favorable prognoses and low proliferative signatures.

In our pathway analysis, we identified distinct E2F activities within essential gene-classified clusters, and 13 of 36 essential genes directly belonged to E2F targets. Specifically, we identified that E2F2/3/4 were activated in cluster 1 patients. The E2F family consists of eight members (E2F1~8), and they are TFs that are responsible for inducing G_1_/S and G_2_/M phase transitions [17,18,19]. It was indicated that E2F2/3 exhibit higher expressions in LUAD tumors compared to healthy lung tissues [20]. E2F2/4 were correlated with LUAD stages and negatively associated with relapse-free survival [20]. Functionally, E2F2 was identified as a direct microRNA (miR)-99a target and is involved in miR-99a- suppression of lung cancer stemness and the epithelial-to-mesenchymal transition [21]. Suppression of E2F3 was shown to synergize the cytotoxicity of paclitaxel in lung cancer cells [22]. However, few studies reported the function of E2F4 in LUAD. Hence, future studies to achieve a more detailed understanding of E2F4 are still needed. Interestingly, in the present study, we identified that E2F2, E2F3, and E2F4 were not essential genes in lung cancer cells from Project Achilles. This result suggests that E2F members have redundant roles, and knocking-out one of the E2F members might not sufficiently reduce essential gene expressions. Still, future experimental studies are needed to verify this speculation.

In the copy number variation analyses, amplification of chromosome 3q26.2 was enriched in cluster 1 patients (48%). The *protein kinase C iota* (*PRKCI*) gene, located in chromosome 3q26.2, was indicated to phosphorylate the cancer stemness regulator, SOX2 [23]. PRKCI/SOX2 signaling promotes the Hedgehog pathway and sustains lung squamous cell cancer stemness [23]. However, few studies have investigated the roles of chromosome 3q26.2 amplification and PRKCI function in LUAD. Gene mutation analyses revealed that non-silent somatic mutations of *TP53* were highly enriched in cluster 1 cancer patients. *TP53* is a well-known tumor suppressor, and it is frequently mutated in various cancers including LUAD. *TP53* functions as a cell cycle suppressor by promoting G_1_/S and G_2_/M arrest [24]. Thus, the loss of function of *TP53* leads to uncontrolled proliferative features of lung cancers. Moreover, one study reported that *TP53* suppresses transcription activities of E2F proteins [25]. A pan-cancer analysis of genomic profiles of patients indicated that E2F signaling is activated in *TP53*-mutant tumors [26]. Those previous results echo our findings that activation of E2F signaling accompanies *TP53* mutations. However, 24% of LUAD patients in cluster 1 harbored tumors with WT *TP53*. No significant differences in E2F activities between WT and mutant *TP53* were found in cluster 1 (Wilcox *p* = 0.1794, data not shown). These findings might be explained by other non-mutational mechanisms suppressing *TP53*’s functions, since MDM2, MDM4, and PPM1D were identified to function as negative regulators of *TP53* [27,28,29]. It is worth noting that *MDM2* gene expression was significantly higher with WT *TP53* compared to the mutant type (Wilcox *p* = 0.03829, data not shown) in cluster 1. Taken together, these findings imply that upstream genomic mutations might not comprehensively capture clinically distinct patients, since alternative signaling might compensate for their roles. In contrast, cancer essential genes which are predominately involved in direct processes of the cell cycle can more accurately reflect clinical differences in LUAD patients.

Immune infiltration and the TMB are crucial factors determining the efficacy of immunotherapy. Our data demonstrated significant differences in the mutation burden within essential gene-classified clusters. Cluster 1 patients had the highest mutation burden. Because *TP53* maintains genomic stability [30], loss of function of *TP53* accompanied by activation of cell proliferative signatures might lead to genomic instability and the high TMB in cluster 1 patients. No significant differences in immune cell infiltration within essential gene-classified clusters were identified, since these essential genes were derived from cancer cell line knockout experiments without considering the tumor microenvironment. In contrast, previously reported expression subtypes were demonstrated to possess distinct immune infiltration levels [31], but they did not exhibit prominent differences in E2F signaling activities.

In our analysis, we identified that essential genes expression exhibited an increased trend in an order of cluster 3, 2 and 1 judging from the heatmap from Figure 2E, Appendix A. Further, a strong association among these essential genes implicated that it is feasible to reduce the genetic signature and develop a simplifier prediction model. In the future, a functional study is needed to better characterize the roles of essential genes in LUAD. This could further guide us to categorize those genes into functional distinct subsets. Then, we could select more representative genes from those subsets to refine our predictive model.

## 5. Conclusions

An integration of our classification with previously reported subtypes provides a more-thorough understanding of immune and proliferative profiles of LUAD. The TRU subtype with high immune infiltration could be further divided into low and high proliferative groups based on our identified essential genes. Consequently, these findings can guide us in identifying subgroups of LUAD tumors that may be vulnerable to immunotherapy or cell proliferation inhibitors in the future.

## Figures and Tables

**Figure 1 cancers-13-02128-f001:**
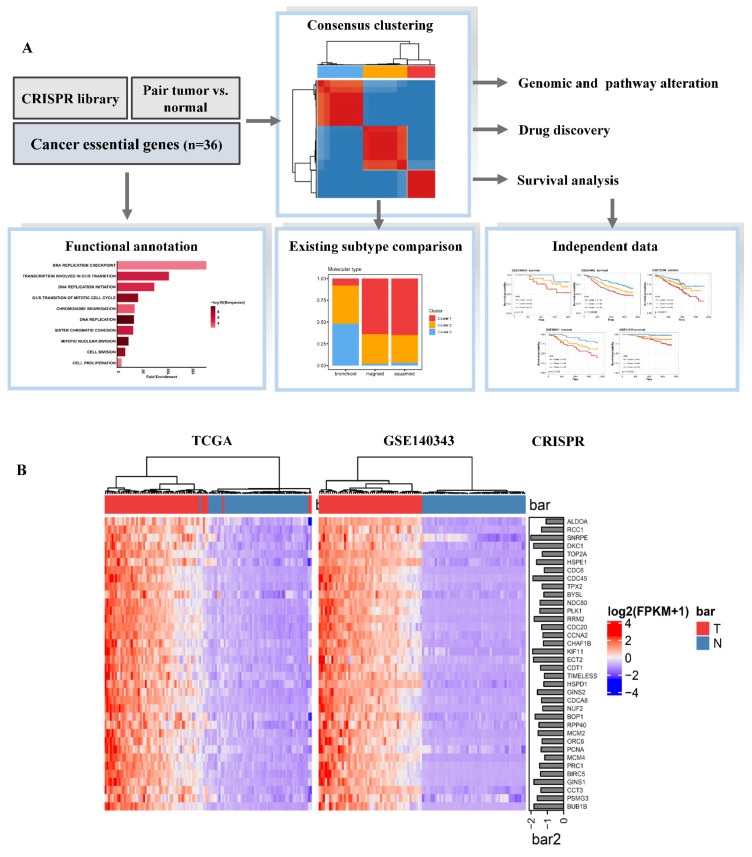
Identification of cancer essential genes in lung adenocarcinoma (LUAD). (**A**) Flowchart demonstrating our investigation of the clinical importance and molecular associations among LUAD essential gene-stratified clusters. (**B**) Heatmap showing significantly upregulated genes in LUAD tumor tissues compared to paired normal tissues using RNA sequencing data of TCGA and GSE140343. The horizontal bar plot represents the median CERES score in LUAD cells from Project Achilles. (**C**) Associations of cancer essential genes from TCGA data are shown as a correlation heatmap. (**D**) Horizontal bar plot demonstrating the top essential gene-enriched signaling pathways using pathway enrichment analyses.

**Figure 2 cancers-13-02128-f002:**
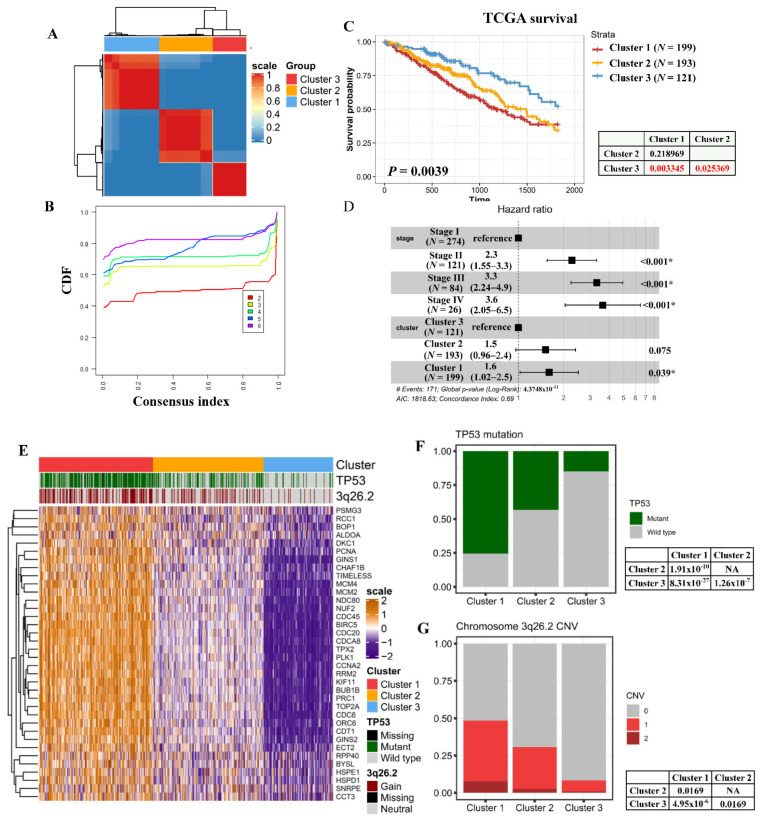
Essential gene-stratified clusters exhibit distinct survival and molecular profiles. (**A**) Similarity matrix of TCGA patients derived from consensus clustering assays shown as a heatmap. (**B**) Cumulative distribution function (CDF) plot used to decide optimal cluster numbers. (**C**) Survival differences within each cluster compared using a log-rank test and demonstrated as a Kaplan-Meier plot. (**D**) Forest plot indicating hazard ratios of essential gene-stratified clusters considering tumor stages. (**E**) Heatmap indicating expressions of essential genes within clusters. The status of the *TP53* mutation and chromosome 3q26.2 alterations are demonstrated as annotation plots. The frequency of *TP53* mutations (**F**) and chromosome 3q26.2 alterations (**G**) within each cluster are demonstrated. (**H**) Violin plots demonstrating the top signaling and transcription factor (TF) activities that significantly differed within essential gene-classified clusters. * means *p* < 0.05.

**Figure 3 cancers-13-02128-f003:**
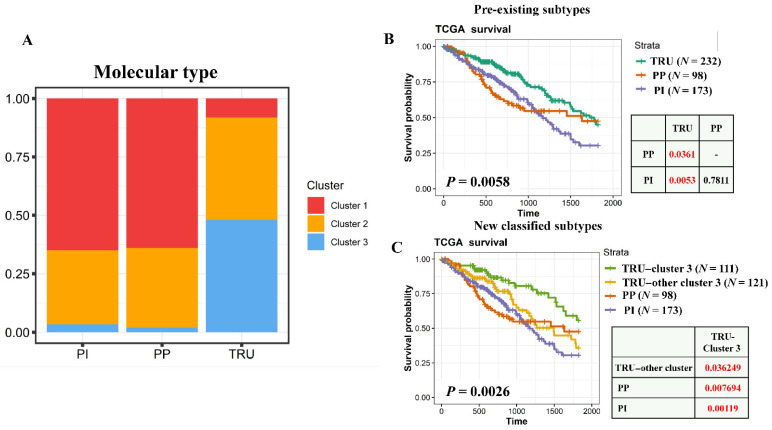
Essential gene-stratified clusters identified a subset from terminal respiratory unit (TRU) tumors with distinct proliferative signatures and prognoses. (**A**) Bar plot showing the frequencies of essential gene-classified clusters in previously defined molecular types. Survival differences within previously defined molecular types (**B**) and essential gene-classified clusters (**C**) are demonstrated as Kaplan-Meier plots using log-rank tests.

**Figure 4 cancers-13-02128-f004:**
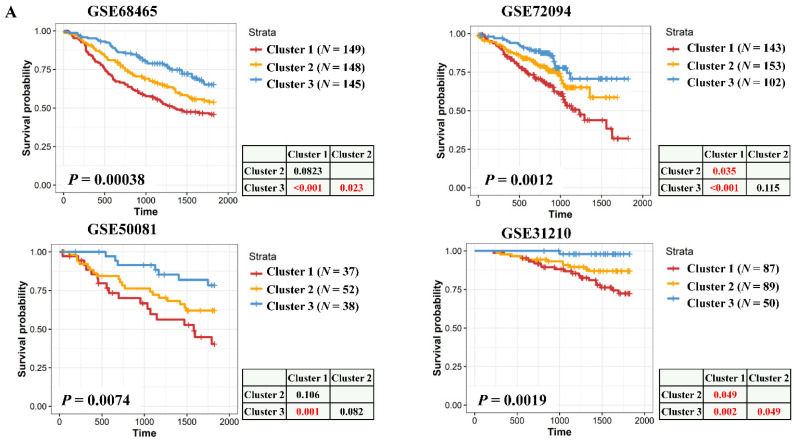
Validation of survival and molecular differences of essential gene-stratified clusters of multiple GEO data. (**A**) Kaplan-Meier plots demonstrating survival differences of essential gene-classified clusters in the GSE140343, GSE68465, GSE72094, GSE50081, and GSE31210 datasets. Survival differences were evaluated using log-rank tests. (**B**) Forest plot using multivariate Cox regression analyses showing essential gene-classified clusters as an independent prognostic factor in the GSE68465, GSE72094, GSE50081, and GSE31320 datasets. (**C**) Distributions of essential gene-classified clusters in previously reported expression subtypes shown in stack bar plots. Kaplan-Meier plots demonstrated survival rates of different molecular types with or without considering cancer essential gene-classified clusters. * means *p* < 0.05.

**Figure 5 cancers-13-02128-f005:**
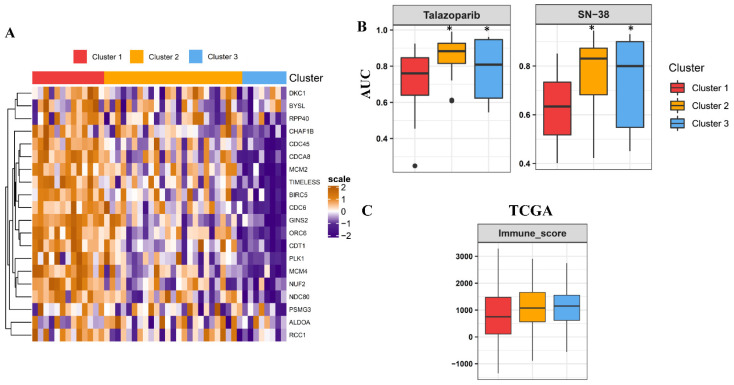
Drug discovery and immune characterization in essential gene-classified clusters of lung adenocarcinoma (LUAD). (**A**) Heatmap showing essential gene-stratified-clusters exhibiting distinct expressions in GDSC lung cancer cells. (**B**) Drug response area under the receiver operator characteristics curve (AUC) of SN-38 and talazoparib within essential gene-classified clusters demonstrated as a boxplot. ESTIMATE-derived immune infiltration (**C**) and the tumor mutation burden (**D**) were compared within each cluster and are shown as boxplots Boxplot of immune infiltration and E2F signaling within previously defined subgroups (**E**) and essential gene-classified clusters (**F**). (**G**) Integration of previously defined molecular types and essential gene-stratified clusters categorizing LUAD patients into distinct immune infiltration and proliferative signature groups. * means *p* < 0.05.

## Data Availability

The data presented in this study are available on request from the corresponding author. The data are not publicly available due to privacy concerns.

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
