# Peer review of "Cancer Essential Genes Stratified Lung Adenocarcinoma Patients with Distinct Survival Outcomes and Identified a Subgroup from the Terminal Respiratory Unit Type with Different Proliferative Signatures in Multiple Cohorts"

_cancers, 2021, doi:10.3390/cancers13092128_

Round 1

Reviewer 1 Report

This work identified 36 cancer essential genes for lung adenocarcinoma by overlapping tumor/normal differentially expressed in tissue samples and CRISPR screening in cancer cell lines. The clusters from these 36 genes are correlated with patient survival, TP53 mutations status, E2F target activities, and tumor mutation burden. The design is well thought and analyses are comprehensive. The writing is clear and easy to understand. I have following questions and comments for the authors to address.

  • Cancer critical genes are more likely related to cell survival and proliferation (which is demonstrated). On tumor phenotypes, they may be responsible for tumor aggressiveness or grade of differentiation. Did you authors look if clusters from these genes are correlated with tumor grade? If you adjust tumor grade in your models, do they still have good prediction?
  • In the potential drug selection, 5 drugs were tested but not reason was given. The authors may want to screen other drugs and see if there are better ones than the selected.
  • No description is given for the GDSC data analysis. Are lung cancer cells all from adenocarcinoma? How are the clusters derived? Cluster 2 and 3 appear no difference?
  • In practice, it is not feasible to do cluster type of classification. In general, these essential genes appear higher in cluster 1 than cluster 2, and cluster 2 than cluster 3. Is this true? Since they are highly correlated, is there a way to find a subset of more independent genes and develop a simple classifier? This needs not to be part of this work but some initial impression or discussion is appreciated.

Reviewer 2 Report

The authors (Ho et al.) have investigated to identify cancer essential genes for lung adenocarcinoma (LUAD) stratification and their clinical and biological differences using multiple cohorts, and they have identified a subgroup from the terminal respiratory unit type with different proliferative signatures based on the LUAD essential genes they found.

Their study is interesting.  However, there are some weaknesses as described bellows.

  1. Subtype classification of the LUAD is pathologically performed in routine pathological diagnosis.The authors should investigate whether cluster 1-3 are associated with pathological subclassification of LUAD.  Also, they should investigate whether TRU-cluster 3 group is associated with pathological subclassification of LUAD.  I suspect that minimally invasive adenocarcinoma and/or lepidic adenocarcinoma are enriched in TRU-cluster 3 group.

  1. Please provide the data of 31 L|UAD cell lines used to identify 693 genes crucial for maintaining survival in supplementary information.

  1. In Figure 2G “chromosome 3q26.3 CNV” is used and “3q26.2” is used in the main text and figure legend.Which is correct?

  1. Font size of Figures 2D and 4B are too small. Please revise them.

  1. Figure 2E: Wildetype -> wild type.

Round 2

Reviewer 1 Report

I think the authors might have mixed tumor grade of differentiation (the question was really asked) with tumor stage. They may be correlated but are different. Tumor grade is a very good prognostic factor for some cancers although it is not that consistent for lung cancer.  It is good the authors conducted more analyses with tumor stage, the major predictor.

Reviewer 2 Report

The revision by the authors (Ho et al.) have been correctly done in comments #1-#4.  However, in comment #5, the revision is not sufficiently.  Please revise “wilde type” into “Wild type”.
